# TEMPERATURE SCHEDULES FOR SELF-SUPERVISED CONTRASTIVE METHODS ON LONG-TAIL DATA

**Anna Kukleva**[*1]**, Moritz Böhle**[*1]**, Bernt Schiele**[1]**, Hilde Kuehne**[2,3]**, Christian Rupprecht**[4]

[1] MPI for Informatics, Saarland Informatics Campus, [2] Goethe University Frankfurt,
[3] MIT-IBM Watson AI Lab, [4] University of Oxford ‖ {akukleva,mboehle}@mpi-inf.mpg.de

## ABSTRACT

Most approaches for self-supervised learning (SSL) are optimised on curated balanced datasets, *e.g.* ImageNet, despite the fact that natural data usually exhibits long-tail distributions. In this paper, we analyse the behaviour of one of the most popular variants of SSL, *i.e.* contrastive methods, on long-tail data. In particular, we investigate the role of the temperature parameter $\tau$ in the contrastive loss, by analysing the loss through the lens of average distance maximisation, and find that a large $\tau$ emphasises group-wise discrimination, whereas a small $\tau$ leads to a higher degree of instance discrimination. While $\tau$ has thus far been treated exclusively as a *constant* hyperparameter, in this work, we propose to employ a *dynamic* $\tau$ and show that a simple cosine schedule can yield significant improvements in the learnt representations. Such a schedule results in a constant 'task switching' between an emphasis on instance discrimination and group-wise discrimination and thereby ensures that the model learns both group-wise features, as well as instance-specific details. Since frequent classes benefit from the former, while infrequent classes require the latter, we find this method to consistently improve separation between the classes in long-tail data without any additional computational cost.

## 1 INTRODUCTION

Deep Neural Networks have shown remarkable capabilities at learning representations of their inputs that are useful for a variety of tasks. Especially since the advent of recent self-supervised learning (SSL) techniques, rapid progress towards learning universally useful representations has been made.

Currently, however, SSL on images is mainly carried out on benchmark datasets that have been constructed and curated for supervised learning (*e.g.* ImageNet (Deng et al., 2009), CIFAR (Krizhevsky et al., 2009), etc.). Although the labels of curated datasets are not *explicitly* used in SSL, the *structure* of the data still follows the predefined set of classes. In particular, the class-balanced nature of curated datasets could result in a learning signal for unsupervised methods. As such, these methods are often not evaluated in the settings they were designed for, *i.e.* learning from truly unlabelled data. Moreover, some methods (*e.g.* (Asano et al., 2019; Caron et al., 2020)) even explicitly enforce a uniform prior over the embedding or label space, which cannot be expected to hold for uncurated datasets.

In particular, uncurated, real-world data tends to follow long-tail distributions (Reed, 2001), in this paper, we analyse SSL methods on long-tailed data. Specifically, we analyse the behaviour of contrastive learning (CL) methods, which are among the most popular learning paradigms for SSL.

In CL, the models are trained such that embeddings of different samples are repelled, while embeddings of different 'views' (*i.e.* augmentations) of the same sample are attracted. The strength of those attractive and repelling forces between samples is controlled by a temperature parameter $\tau$, which has been shown to play a crucial role in learning good representations (Chen et al., 2020c;a). To the best of our knowledge, $\tau$ has thus far almost exclusively been treated as a *constant* hyper-parameter.

In contrast, we employ a *dynamic* $\tau$ during training and show that this has a strong effect on the learned embedding space for long-tail distributions. In particular, by introducing a simple schedule for $\tau$ we consistently improve the representation quality across a wide range of settings. Crucially, these gains are obtained without additional costs and only require oscillating $\tau$ with a cosine schedule.

---

[*]equal contribution. Code available at: github.com/annusha/temperature_schedules

This mechanism is grounded in our novel understanding of the effect of temperature on the contrastive loss. In particular, we analyse the contrastive loss from an average distance maximisation perspective, which gives intuitive insights as to why a large temperature emphasises *group-wise discrimination*, whereas a small temperature leads to a higher degree of *instance discrimination* and more uniform distributions over the embedding space. Varying $\tau$ during training ensures that the model learns both group-wise and instance-specific features, resulting in better separation between head and tail classes.

Overall, our contributions are summarised as follows: • we carry out an extensive analysis of the effect of $\tau$ on imbalanced data; • we analyse the contrastive loss from an average distance perspective to understand the emergence of semantic structure; • we propose a simple yet effective temperature schedule that improves the performance across different settings; • we show that the proposed $\tau$ scheduling is robust and consistently improves the performance for different hyperparameter choices.

## 2 RELATED WORK

Self-supervised representation learning (SSL) from visual data is a quickly evolving field. Recent methods are based on various forms of comparing embeddings between transformations of input images. We divide current methods into two categories: contrastive learning (He et al., 2020; Chen et al., 2020c;a; Oord et al., 2018), and non-contrastive learning (Grill et al., 2020; Zbontar et al., 2021; Chen & He, 2021; Bardes et al., 2022; Wei et al., 2022; Gidaris et al., 2021; Asano et al., 2019; Caron et al., 2020; He et al., 2022). Our analysis concerns the structure and the properties of the embedding space of contrastive methods when training on imbalanced data. Consequently, this section focuses on contrastive learning methods, their analysis and application to imbalanced training datasets.

**Contrastive Learning** employs instance discrimination (Wu et al., 2018) to learn representations by forming positive pairs of images through augmentations and a loss formulation that maximises their similarity while simultaneously minimising the similarity to other samples. Methods such as MoCo (He et al., 2020; Chen et al., 2020c), SimCLR (Chen et al., 2020a;b), SwAV (Caron et al., 2020), CPC (Oord et al., 2018), CMC Tian et al. (2020a), and Whitening (Ermolov et al., 2021) have shown impressive representation quality and down-stream performance using this learning paradigm. CL has also found applications beyond SSL pre-training, such as multi-modal learning (Shvetsova et al., 2022), domain generalisation (Yao et al., 2022), semantic segmentation (Van Gansbeke et al., 2021), 3D point cloud understanding (Afham et al., 2022), and 3D face generation (Deng et al., 2020).

**Negatives.** The importance of negatives for contrastive learning is remarkable and noticed in many prior works (Wang et al., 2021; Yeh et al., 2021; Zhang et al., 2022; Iscen et al., 2018; Kalantidis et al., 2020; Robinson et al., 2020; Khaertdinov et al., 2022). Yeh et al. (2021) propose decoupled learning by removing the positive term from the denominator, Robinson et al. (2020) develop an unsupervised hard-negative sampling technique, Wang et al. (2021) propose to employ a triplet loss, and Zhang et al. (2022); Khaertdinov et al. (2022) propose to improve negative mining with the help of different temperatures for positive and negative samples that can be defined as input-independent or input-dependent functions, respectively. In contrast to *explicitly* choosing a specific subset of negatives, we discuss the Info-NCE loss (Oord et al., 2018) through the lens of an average distance perspective with respect to all negatives and show that the temperature parameter can be used to *implicitly* control the effective number of negatives.

**Imbalanced Self-Supervised Learning.** Learning on imbalanced data instead of curated balanced datasets is an important application since natural data commonly follows long-tailed distributions (Reed, 2001; Liu et al., 2019; Wang et al., 2017). In recent work, Kang et al. (2020), Yang & Xu (2020), Liu et al. (2021), Zhong et al. (2022), Gwilliam & Shrivastava (2022) discover that self-supervised learning generally allows to learn a more robust embedding space than a supervised counterpart. Tian et al. (2021) explore the down-stream performance of contrastive learning on standard benchmarks based on large-scale uncurated pre-training and propose a multi-stage distillation framework to overcome the shift in the distribution of image classes. Jiang et al. (2021); Zhou et al. (2022) propose to address the data imbalance by identifying and then emphasising tail samples during training in an unsupervised manner. For this, Jiang et al. (2021) compare the outputs of the trained model before and after pruning, assuming that tail samples are more easily 'forgotten' by the pruned model and can thus be identified. Zhou et al. (2022), use the loss value for each input to identify tail samples and then use stronger augmentations for those. Instead of modifying the architecture

or the training data of the underlying frameworks, we show that a simple approach—*i.e.* oscillating the temperature of the Info-NCE loss (Oord et al., 2018) to alternate between instance and group discrimination—can achieve similar performance improvements at a low cost.

**Analysis of Contrastive Learning (CL).** Given the success of CL in representation learning, it is essential to understand its properties. While some work analyses the interpretability of embedding spaces (Bau et al., 2017; Fong & Vedaldi, 2018; Laina et al., 2020; 2021), here the focus lies on understanding the structure and learning dynamics of the objective function such as in Saunshi et al. (2019); Tsai et al. (2020); Chen et al. (2021). E.g., Chen et al. (2021) study the role of the projection head, the impact of multi-object images, and a feature suppression phenomenon. Wen & Li (2021) analyse the feature learning process to understand the role of augmentations in CL. Robinson et al. (2021) find that an emphasis on instance discrimination can improve representation of some features at the cost of suppressing otherwise well-learned features. Wang & Isola (2020); Wang & Liu (2021) analyse the uniformity of the representations learned with CL. In particular, Wang & Liu (2021) focus on the impact of individual negatives and describe a uniformity-tolerance dilemma when choosing the temperature parameter. In this work, we rely on the previous findings, expand them to long-tailed data distributions and complement them with an understanding of the emergence of semantic structure.

## 3    METHOD

In the following, we describe our approach and analysis of contrastive learning on long-tailed data. For this, we will first review the core principles of contrastive learning for the case of uniform data (Sec. 3.1). In Sec. 3.2, we then place a particular focus on the temperature parameter $\tau$ in the contrastive loss and its impact on the learnt representations. Based on our analysis, in Sec. 3.3 we discuss how the choice of $\tau$ might negatively affect the learnt representation of rare classes in the case of long-tailed distributions. Following this, we describe a simple proof-of-concept based on additional coarse supervision to test our hypothesis. We then further develop temperature schedules (TS) that yield significant gains with respect to the separability of the learnt representations in Sec. 4.

### 3.1    CONTRASTIVE LEARNING

**The Info-NCE loss** is a popular objective for contrastive learning (CL) and has lead to impressive results for learning useful representations from unlabelled data (Oord et al., 2018; Wu et al., 2018; He et al., 2020; Chen et al., 2020a). Given a set of inputs $\{x_1, \ldots, x_N\}$, and the cosine similarities $s_{ij}$ between learnt representations $u_i = f(\mathcal{A}(x_i))$ and $v_j = g(\mathcal{A}(x_j))$ of the inputs, the loss is defined by:

$$\mathcal{L}_c = \sum_{i=1}^{N} -\log \frac{\exp(s_{ii}/\tau)}{\exp(s_{ii}/\tau) + \sum_{j \neq i} \exp(s_{ij}/\tau)}. \tag{1}$$

Here, $\mathcal{A}(\cdot)$ applies a random augmentation to its input and $f$ and $g$ are deep neural networks. For a given $x_i$, we will refer to $u_i$ as the *anchor* and to $v_j$ as a *positive* sample if $i = j$ and as a *negative* if $i \neq j$. Last, $\tau$ denotes the *temperature* of the Info-NCE loss and has been found to crucially impact the learnt representations of the model (Wang & Isola, 2020; Wang & Liu, 2021; Robinson et al., 2021).

**Uniformity.** Specifically, a small $\tau$ has been tied to more uniformly distributed representations, see Fig. 1. For example, Wang & Liu (2021) show that the loss is 'hardness-aware', *i.e.* negative samples closest to the anchor receive the highest gradient. In particular, for a given anchor, the gradient with respect to the negative sample $v_j$ is scaled by its relative contribution to the denominator in Eq. (1):

$$\frac{\partial \mathcal{L}_c}{\partial v_j} = \frac{\partial \mathcal{L}_c}{\partial s_{ij}} \times \frac{\partial s_{ij}}{\partial v_j} = \frac{1}{\tau} \times [\mathrm{softmax}_k(s_{ik}/\tau)]_j \times \frac{\partial s_{ij}}{\partial v_j} \quad . \tag{2}$$

As a result, for sufficiently small $\tau$, the model minimises the cosine similarity to the nearest negatives in the embedding space, as softmax approaches an indicator function that selects the largest gradient. The optimum of this objective, in turn, is to distribute the embeddings as uniformly as possible over the sphere, as this reduces the average similarity between nearest neighbours, see also Figs. 1 and 3.

**Semantic structure.** In contrast, a large $\tau$ has been observed to induce more semantic structure in the representation space. However, while the effect of small $\tau$ has an intuitive explanation, the phenomenon that larger $\tau$ induce semantic structure is much more poorly understood and has mostly

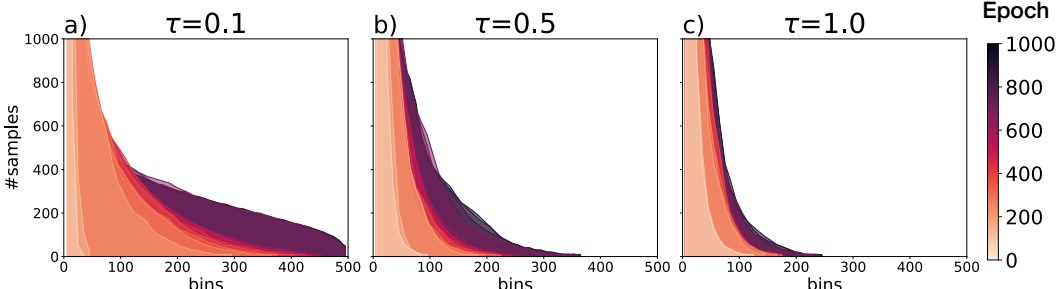

Figure 1: **Coverage of the embedding space during training.** To measure coverage we uniformly sample 500 bins on the unit sphere. Each training sample is assigned to the closest bin and we plot a histogram of the assignments. X-axis: bins. Y-axis: number of training samples in a bin. Colors denotes epochs: light is the 1st epoch of training, dark is the last. For small $\tau$ (a) the representations are more uniformly distributed (cf. Sec. 3).

been described empirically (Wang & Liu, 2021; Robinson et al., 2021). Specifically, note that for any given positive sample, all negatives are repelled from the anchor, with close-by samples receiving exponentially higher gradients. Nonetheless, for large $\tau$, tightly packed semantic clusters emerge. However, if close-by negatives are heavily repelled, how can this be? Should the loss not be dominated by the hard-negative samples and thus break the semantic structure?

To better understand both phenomena, we propose to view the contrastive loss through the lens of *average distance* maximisation, which we describe in the following section.

## 3.2 CONTRASTIVE LEARNING AS AVERAGE DISTANCE MAXIMISATION

As discussed in the previous section, the parameter $\tau$ plays a crucial role in shaping the learning dynamics of contrastive learning. To understand this role better, in this section, we present a novel viewpoint on the mechanics of the contrastive loss that explain the observed model behaviour. In particular, and in contrast to Wang & Liu (2021) who focused on the impact of *individual* negatives, for this we discuss the *cumulative* impact that all negative samples have on the loss.

To do so, we express the summands $\mathcal{L}_c^i$ of the loss in terms of distances $d_{ij}$ instead of similarities $s_{ij}$:

$$0 \leq d_{ij} = \frac{1 - s_{ij}}{\tau} \leq \frac{2}{\tau} \quad \text{and} \quad c_{ii} = \exp(d_{ii}). \tag{3}$$

This allows us to rewrite the loss $\mathcal{L}_c^i$ as

$$\mathcal{L}_c^i = -\log\left(\frac{\exp\left(-d_{ii}\right)}{\exp\left(-d_{ii}\right) + \sum_{j \neq i} \exp\left(-d_{ij}\right)}\right) = \log\left(1 + c_{ii} \sum_{j \neq i} \exp\left(-d_{ij}\right)\right). \tag{4}$$

As the effect $c_{ii}$ of the positive sample for a given anchor is the same for all negatives, in the following we place a particular focus on the negatives and their relative influence on the loss in Eq. (4); for a discussion of the influence of positive samples, please see appendix A.4.

To understand the impact of the temperature $\tau$, first note that the loss monotonically increases with the sum $S_i = \sum_{j \neq i} \exp(-d_{ij})$ of exponential distances in Eq. (4). As $\log$ is a continuous, monotonic function, we base the following discussion on the impact of $\tau$ on the sum $S_i$.

**For small** $\tau$, the nearest neighbours of the anchor point dominate $S_i$, as differences in similarity are amplified. As a result, the contrastive objective maximises the average distance to nearest neighbours, leading to a uniform distribution over the hypersphere, see Fig. 3. Since individual negatives dominate the loss, this argument is consistent with existing interpretations, *e.g.* Wang & Liu (2021), as described in the previous section.

**For large** $\tau$, (*e.g.* $\tau \geq 1$), on the other hand, the contributions to the loss from a given negative are on the same order of magnitude for a wide range of cosine similarities. Hence, the constrastive objective can be thought of as maximising the average distance over a wider range of neighbours. Interestingly, since distant negatives will typically outnumber close negatives, the strongest *cumulative* contribution

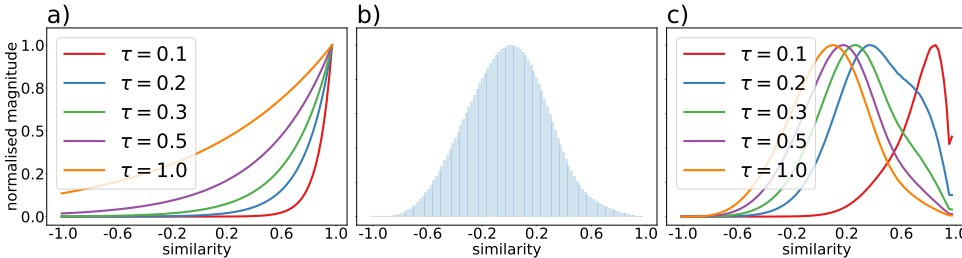

Figure 2: **Loss contribution by similarity.** X-axis: cosine similarity between anchor and negative. All curves are normalised such that their max y-value is 1. **a)**: influence of an individual negative sample to the loss depending on its similarity to anchor for different $\tau$; **b)**: average histogram of distribution of negatives over the hypersphere with respect to their similarity to the anchor; **c)**: cumulative impact that negative samples have on the loss. The *cumulative* contribution of negatives shifts left, towards less similar samples, in contrast to individual contributions of negatives. As $\tau \to \infty$, the cumulative distribution coincides with the histogram b).

to the contrastive loss will come from more distant samples, despite the fact that *individually* the strongest contributions will come from the closest samples. To visualise this, in Fig. 2a, we plot the contributions of *individual* samples depending on their distance, as well as the distribution of similarities $s_{ij}$ to negatives over the entire dataset in Fig. 2b. Since the number of negatives at larger distances (*e.g.* $s_{ij} \approx 0.1$) significantly outnumber close negatives ($s_{ij} > 0.9$), the peak of the cumulative contributions[1] shifts towards lower similarities for larger $\tau$, as can be seen in Fig. 2c; in fact, for $\tau \to \infty$, the distribution of cumulative contributions approaches the distribution of negatives.

Hence, the model can significantly decrease the loss by increasing the distance to relatively 'easy negatives' for much longer during training, *i.e.* to samples that are easily distinguishable from the anchor by simple patterns. Instead of learning 'hard' features that allow for better *instance discrimination* between hard negatives, the model will be biased to learn easy patterns that allow for *group-wise discrimination* and thereby increase the margin between clusters of samples. Note that since the clusters as a whole mutually repel each other, the model is optimised to find a trade-off between the expanding forces between hard negatives (*i.e.* within a cluster) and the compressing forces that arise due to the margin maximisation between easy negatives (*i.e.* between clusters).

Importantly, such a bias towards easy features can prevent the models from learning hard features—*i.e.* by focusing on *group-wise discrimination*, the model becomes agnostic to instance-specific features that would allow for a better *instance discrimination* (cf. Robinson et al. (2021)). In the following, we discuss how this might negatively impact rare classes in long-tailed distributions.

### 3.3 TEMPERATURE SCHEDULES FOR CONTRASTIVE LEARNING ON LONG-TAIL DATA

As discussed in Sec. 1, naturally occurring data typically exhibit long-tail distributions, with some classes occurring much more frequently than others; across the dataset, *head* classes appear frequently, whereas *tail* classes contain fewest number of samples. Since self-supervised learning methods are designed to learn representations from unlabelled data, it is important to investigate their performance on imbalanced datasets.

**Claim: Tail classes benefit from instance discrimination.** As discussed in Sec. 3.2, sufficiently large $\tau$ are required for semantic groups to emerge during contrastive learning as this emphasises group-wise discrimination. However, as shown by Robinson et al. (2021), this can come at the cost of encoding instance-specific features and thus hurt the models' instance discrimination capabilities.

We hypothesise that this disproportionately affects tail classes, as tail classes consist of only relatively few instances to begin with. Their representations should thus *remain distinguishable* from most of their neighbours and not be grouped with other instances, which are likely of a different class. In contrast, since head classes are represented by many samples, grouping those will be advantageous.

To test this hypothesis, we propose to explicitly train head and tail classes with different $\tau$, to emphasise group discrimination for the former while ensuring instance discrimination for the latter.

---

[1]To obtain the cumulative contributions, we group the negatives into 100 non-overlapping bins of size 0.02 depending on their distance to the anchor and report the sum of contributions of a given bin.

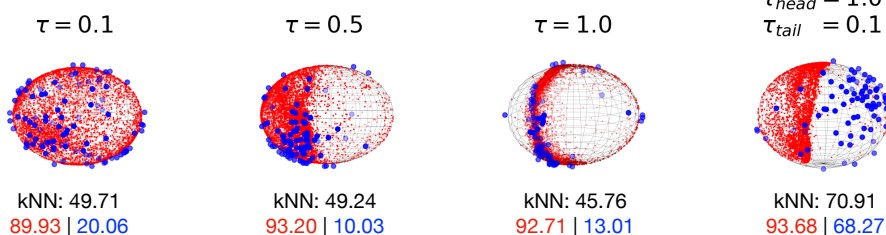

Figure 3: **Representations of a head and a tail class.** Visualisation of the influence of $\tau$ on representations of two semantically close classes (trained with all 10 classes). Red: single head class and blue: single tail class from CIFAR10-LT. Small $\tau=0.1$ promotes uniformity, while large $\tau=1.0$ creates dense clusters. With $\tau_{\{head/tail\}}$ we refer to coarse supervision described in Sec. 3.3 which separates tail from head classes. In black / red / blue, we respectively show the average kNN accuracy over all classes / the head class / the tail class.

**Experiment: Controlling $\tau$ with coarse supervision.** We experiment on CIFAR10-LT (a long-tail variant of CIFAR10 - see Sec. 4.1) in which we select a different $\tau$ depending on whether the anchor $u_i$ is from a head or a tail class, *i.e.* of the 5 *most* or *least* common classes. We chose a relatively large $\tau$ ($\tau_{\text{head}}=1.0$) for the 5 head classes to emphasise group-wise discrimination and a relatively small $\tau$ ($\tau_{\text{tail}}=0.1$) for the 5 tail classes to encourage the model to learn instance-discriminating features.

As can be seen in Fig. 3, this simple manipulation of the contrastive loss indeed provides a significant benefit with respect to the semantic structure of the embedding space, despite only weakly supervising the learning by adjusting $\tau$ according to a coarse (frequent/infrequent) measure of class frequency.

In particular, in Fig. 3, we show the projections of a single head class and a single tail class onto the three leading PCA dimensions and the corresponding kNN accuracies. We would like to highlight the following results. First, without any supervision, we indeed find that the head class consistently performs better for larger values of $\tau$ (*e.g.* 1.0), whereas the tail class consistently benefits from smaller values for $\tau$ (*e.g.* 0.1). Second, when training the model according to the coarse $\tau$ supervision as described above, we are not only able to maintain the benefits of large $\tau$ values for the head class, but significantly outperform all constant $\tau$ versions for the tail class, which improves the overall model performance on all classes; detailed results for all classes are provided in the appendix.

**Temperature Schedules (TS) without supervision.** Such supervision with respect to the class frequency is, of course, generally not available when training on unlabelled data and these experiments are only designed to test the above claim and provide an intuition about the learning dynamics on long-tail data. However, we would like to point out that the supervision in these experiments is very coarse and only separates the unlabelled data into *frequent* and *infrequent* classes. Nonetheless, while the results are encouraging, they are, of course, based on additional, albeit coarse, labels. Therefore, in what follows, we present an unsupervised method that yields similar benefits.

In detail, we propose to modify $\tau$ according to a cosine schedule, such that it alternates between an upper ($\tau_+$) and a lower ($\tau_-$) bound at a fixed period length $T$:

$$\tau_{\cos}(t) = (\tau_+ - \tau_-) \times (1 + \cos(2\pi\,t/T))/2 + \tau_- \; ; \tag{5}$$

here, $t$ denotes training epochs. This method is motivated by the observation that $\tau$ controls the trade-off between learning easily separable features and learning instance-specific features.

Arguably, however, the models should learn both types of features: *i.e.* the representation space should be structured according to easily separable features that (optimally) represent semantically meaningful group-wise patterns, whilst still allowing for instance discrimination within those groups.

Therefore, we propose to *alternate* between both objectives as in Eq. (5), to ensure that throughout training the model learns to encode instance-specific patterns, whilst also structuring the representation space along semantically meaningful features. Note that while we find a cosine schedule to work best and to be robust with respect to the choice for $T$ (Sec. 4.3), we also evaluate alternatives. Even randomly sampling $\tau$ from the interval $[\tau_-, \tau_+]$ improves the model performance. This indicates that the *task switching* between group-wise discrimination (large $\tau$) and instance discrimination (small $\tau$) is indeed the driving factor behind the performance improvements we observe.

## 4 EXPERIMENTAL RESULTS

In this section, we validate our hypothesis that simple manipulations of the temperature parameter in Eq. (1) lead to better performance for long-tailed data. First, we introduce our experimental setup in Sec. 4.1, then in Sec. 4.2 we discuss the results across three imbalanced datasets and, finally, we analyse different design choices of the framework through extensive ablation studies in Sec. 4.3.

### 4.1 IMPLEMENTATION DETAILS

**Datasets.** We consider long-tailed (LT) versions of the following three popular datasets for the experiments: CIFAR10-LT, CIFAR100-LT, and ImageNet100-LT. For most of the experiments, we follow the setting from SDCLR (Jiang et al., 2021). In case of **CIFAR10-LT/CIFAR100-LT**, the original datasets (Krizhevsky et al., 2009) consist of 60000 32x32 images sampled uniformly from 10 and 100 semantic classes, respectively, where 50000 images correspond to the training set and 10000 to a test set. Long-tail versions of the datasets are introduced by Cui et al. (2019) and consist of a subset of the original datasets with an exponential decay in the number of images per class. The imbalance ratio controls the uniformity of the dataset and is calculated as the ratio of the sizes of the biggest and the smallest classes. By default, we use an imbalance ratio 100 if not stated otherwise. Experiments in Tab. 1, Tab. 3 are the average over three runs with different permutations of classes. **ImageNet100-LT** is a subset of the original ImageNet-100 (Tian et al., 2020a) consisting of 100 classes for a total of 12.21k 256x256 images. The number of images per class varies from 1280 to 25.

**Training.** We use an SGD optimizer for all experiments with a weight decay of 1e-4. As for the learning rate, we utilize linear warm-up for 10 epochs that is followed by a cosine annealing schedule starting from 0.5. We train for 2000 epochs for CIFAR10-LT and CIFAR100-LT and 800 epochs for ImageNet100-LT. For CIFAR10-LT and CIFAR100-LT we use a ResNet18 (He et al., 2016) backbone. For ImageNet100-LT we use a ResNet50 (He et al., 2016) backbone. For both the MoCo (He et al., 2020) and the SimCLR (Chen et al., 2020a) experiments, we follow Jiang et al. (2021) and use the following augmentations: resized crop, color jitters, grey scale and horizontal flip. MoCo details: we use a dictionary of size 10000, a projection dimensionality of 128 and a projection head with one linear layer. SimCLR details: we train with a batch size of 512 and a projection head that has two layers with an output size of 128. For evaluation, we discard the projection head and apply l2-normalisation. Regarding the proposed temperature schedules (TS), we use a period length of $T{=}400$ with $\tau_+{=}1.0$ and $\tau_-{=}0.1$ if not stated otherwise; for more details, see appendix A.2.

**Evaluation** We use k nearest neighbours (kNN) and linear classifiers to assess the learned features. For kNN, we compute $l2$-normalised distances between LT samples from the train set and the class-balanced test set. For each test image, we assign it to the majority class among the top-k closest train images. We report accuracy for kNN with $k{=}1$ (kNN@1) and with $k{=}10$ (kNN@10). Compared to fine-tuning or linear probing, kNN directly evaluates the learned embedding since it relies on the learned metric and local structure of the space. We also evaluate the linear separability and generalisation of the space with a linear classifier that we train on the top of frozen backbone. For this, we consider two setups: balanced few-shot linear probing (FS LP) and long-tailed linear probing (LT LP). For FS LP, the few-shot train set is a direct subset of the original long-tailed train set with the shot number equal to the minimum class size in the original LT train set. For LT LP, we use the original LT training set. For extended tables, see appendix A.3.

### 4.2 EFFECTIVENESS OF TEMPERATURE SCHEDULES

**Contrastive learning with TS.** In Tab. 1 we present the efficacy of temperature schedules (TS) for two well-known contrastive learning frameworks MoCo (He et al., 2020) and SimCLR (Chen et al., 2020a). We find that both frameworks benefit from varying the temperature and we observe consistent improvements over all evaluation metrics for CIFAR10-LT and CIFAR100-LT, *i.e.* the local structure of the embedding space (kNN) and the global structure (linear probe) are both improved. Moreover, we show in Tab. 3 that our finding also transfers to ImageNet100-LT. Furthermore, in Tab. 2 we evaluate the performance of the proposed method on the CIFAR10 and CIFAR100 datasets with different imbalance ratios. An imbalance ratio of 50 (imb50) reflects less pronounced imbalance, and imb150 corresponds to the datasets with only 30 (CIFAR10) and 3 (CIFAR100) samples for the

smallest class. Varying $\tau$ during training improves the performance for different long-tailed data; for a discussion on the dependence of the improvement on the imbalance ratio, please see the appendix.

| | CIFAR10-LT | | | | CIFAR100-LT | | | |
|---|---|---|---|---|---|---|---|---|
| method | kNN@1 | kNN@10 | FS LP | LT LP | kNN@1 | kNN@10 | FS LP | LT LP |
| MoCo | 63.54 | 64.56 | 69.31 | 65.11 | 28.69 | 28.75 | 26.86 | 30.41 |
| MoCo + TS | **64.99** | **65.01** | **72.87** | **66.86** | **30.31** | **29.75** | **28.97** | **32.05** |
| SimCLR | 59.84 | 60.19 | 68.29 | 61.86 | 28.81 | 28.12 | 25.70 | 31.20 |
| SimCLR + TS | **63.09** | **62.91** | **71.86** | **65.03** | **31.06** | **30.06** | **28.89** | **33.28** |

Table 1: **Effect of temperature scheduling.** Comparison of MoCo vs MoCo+TS and SimCLR vs SimCLR+TS on CIFAR10-LT and CIFAR100-LT with kNN, few-shot and long-tail linear probe (FS LP and LT LP).

| | CIFAR-10-LT | | | | CIFAR-100-LT | | | |
|---|---|---|---|---|---|---|---|---|
| | imb 50 | | imb 150 | | imb 50 | | imb 150 | |
| method | kNN@10 | FS LP | kNN@10 | FS LP | kNN@10 | FS LP | kNN@10 | FS LP |
| MoCo | 69.12 | 74.16 | 59.13 | 65.76 | 32.22 | 33.53 | 25.36 | 22.73 |
| MoCo + TS | **71.49** | **76.37** | **60.83** | **68.59** | **33.24** | **35.03** | **26.75** | **22.78** |

Table 2: **Effect of imbalance ratio.** MoCo vs MoCo+TS on CIFAR10-LT and CIFAR100-LT for imbalance ratio 50 (imb50) and 150 (imb150). Evaluation metrics: kNN classifier and few-shot linear probe (FS LP).

| | CIFAR-10-LT | | | CIFAR-100-LT | | | ImageNet-100-LT | | |
|---|---|---|---|---|---|---|---|---|---|
| method | kNN@10 | FS LP | LS LP | kNN@10 | FS LP | LT LP | kNN@10 | FS LP | LT LP |
| SimCLR | 60.19 | 68.29 | 61.68 | 28.12 | 25.70 | 31.20 | 38.00 | 42.64 | 44.82 |
| SDCLR | 60.74 | 71.03 | 64.99 | 29.22 | 27.28 | **34.23** | 37.36 | 42.74 | 46.40 |
| SimCLR + TS | **62.91** | **71.86** | **65.03** | **30.06** | **28.89** | 33.28 | **38.86** | **45.18** | **47.26** |

Table 3: **Comparison with SDCLR.** SimCLR vs SDCLR vs SimCLR+TS on CIFAR10-LT, CIFAR100-LT, and ImageNet100-LT. Evaluation: kNN classifier, few-shot (FS LP) and long-tail linear probe (LT LP).

**TS vs SDCLR.** Further, we compare our method with SDCLR (Jiang et al., 2021). In SDCLR, SimCLR is modified s.t. the embeddings of the online model are contrasted with those of a pruned version of the same model, which is updated after every epoch. Since the pruning is done by simply masking the pruned weights of the original model, SDCLR requires twice as much memory compared to the original SimCLR and extra computational time to prune the model every epoch. In contrast, our method does not require any changes in the architecture or training. In Tab. 3 we show that this simple approach improves not only over the original SimCLR, but also over SDCLR in most metrics.

## 4.3 ABLATIONS

In this section, we evaluate how the hyperparameters in Eq. (5) can influence the model behaviour.

**Cosine Boundaries.** First, we vary the lower $\tau_-$ and upper $\tau_+$ bounds of $\tau$ for the cosine schedule. In Tab. 4 we assess the performance of MoCo+TS with different $\tau_-$ and $\tau_+$ on CIFAR10 with FS LP. We observe a clear trend that with a wider range of $\tau$ values the performance increases. We attribute this to the ability of the model to learn better 'hard' features with low $\tau$ and improve semantic structure for high $\tau$. Note that 0.07 is the value for $\tau$ in many current contrastive learning methods.

| $\tau_-$ \ $\tau_+$ | 0.2 | 0.3 | 0.4 | 0.5 | 1.0 |
|---|---|---|---|---|---|
| 0.07 | 69.46 | 68.86 | 71.29 | 71.83 | **73.26** |
| 0.1 | 68.17 | 70.34 | 71.25 | 72.31 | 72.87 |
| 0.2 | 68.89 | 69.37 | 70.12 | 69.65 | 71.42 |

Table 4: **Influence of cosine boundaries.** Best performance with the largest difference between $\tau_-$ and $\tau_+$. CIFAR10 with MoCo+TS, evaluating few-shot linear probes (FS LP).

| TS | FS LP |
|---|---|
| fixed | 68.89 |
| step | 70.18 |
| rand | 70.26 |
| oscil | 71.50 |
| cos | **72.31** |

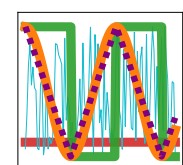

Table 5: **Alternative Schedules**. Constant, step function, and random sampling. All functions are bounded by 0.1 and 0.5.

**Cosine Period.** Further, we investigate if the length of the period $T$ in Eq. (5) impacts the performance of the model. In Tab. 6, we show that modifying the temperature $\tau$ based on the cosine schedule is beneficial during training independently of the period $T$. The performance varies insignificantly depending on $T$ and consistently improves over standard fixed $\tau = 0.2$, whereas the best performance we achieve with $T = 400$. Even though the performance is stable with respect to the length of the period, it changes within one period as we show in Fig. 4. Here, we average the accuracy of one last full period over different models trained with different $T$ and find that the models reach the best performance around $0.7\,T$. Based on this observation, we recommend to stop training after $(n - 0.3)\,T$ epochs, where $n$ is the number of full periods.

| T | T / #epochs | FS LP |
|---|---|---|
| no | fixed $\tau$ | 68.89 |
| 200 | 0.1 | 71.86 |
| 400 | 0.2 | 72.87 |
| 1000 | 0.5 | 72.47 |
| 2000 | 1.0 | 72.22 |
| 4000 | 2.0 | 72.10 |

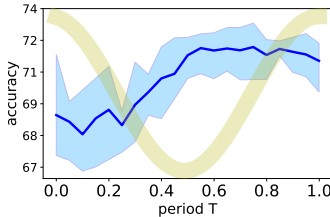

Table 6: **Influence of the period length** $T$. Few-shot linear probe accuracy (FS LP) of MoCo+TS on CIFAR10-LT.

Figure 4: **Dependence on relative time of one period.** Blue: Average FS LP of last period of the models trained with $T = 200, 400, 1000, 2000$. Light blue: variance. Yellow: Relative cosine value over relative time. CIFAR10-LT trained with MoCo+TS.

**Alternatives to Cosine Schedule.** Additionally, we test different methods of varying the temperature parameter $\tau$ and report the results in Tab. 5: we examine a linearly oscillating (oscil) function, a step function, and random sampling. For the linear oscillations, we follow the same schedule as for the cosine version, as shown on the right of Tab. 5. For the step function, we change $\tau$ from a low (0.1) to a high (0.5) value and back every 200 epochs. For random, we uniformly sample values for $\tau$ from the range [0.1, 0.5]. In Tab. 5 we observe that both those methods for varying the $\tau$ value also improve the performance over the fixed temperature, while with the cosine schedule the model achieves the best performance. These results indicate that it is indeed the *task switching* between group-wise and instance-wise discrimination during training which is the driving factor for the observed improvements for unsupervised long-tail representation learning. We assume the reason why slow oscillation of the temperature performs better than fast (*i.e.* random) temperature changes is grounded in learning dynamics and the slow evolution of the embedding space during training.

## 5 CONCLUSION

In this work, we discover the surprising effectiveness of temperature schedules for self-supervised contrastive representation learning on imbalanced datasets. In particular, we find that a simple cosine schedule for $\tau$ consistently improves two state-of-the-art contrastive methods over several datasets and different imbalance ratios, without introducing any additional cost.

Importantly, our approach is based on a novel perspective on the contrastive loss, in which the average distance maximisation aspect is emphasised. This perspective sheds light on which samples dominate the contrastive loss and explains why large values for $\tau$ can lead to the emergence of tight clusters in the embedding space, despite the fact that individual instance *always* repel each other.

Specifically, we find that while a large $\tau$ is thus necessary to induce semantic structure, the concomitant focus on *group-wise* discrimination biases the model to encode easily separable features rather than instance-specific details. However, in long-tailed distributions, this can be particularly harmful to the most infrequent classes, as those require a higher degree of instance discrimination to remain distinguishable from the prevalent semantic categories. The proposed cosine schedule for $\tau$ overcomes this tension, by alternating between an emphasis on instance discrimination (small $\tau$) and group-wise discrimination (large $\tau$). As a result of this constant 'task switching', the model is trained to both structure the embedding space according to semantically meaningful features, whilst also encoding instance-specific details such that rare classes remain distinguishable from dominant ones.

ETHICS STATEMENT

The paper proposes an analysis and a method to improve the performance of self-supervised representation learning methods based on the contrastive loss. The method and investigation in this paper do not introduce any ethical issues to the field of representation learning, as it is decoupled from the training data. Nonetheless, we would like to point out that representation learning does not automatically prevent models from learning harmful biases from the training data and should not be used outside of research applications without thorough evaluation for fairness and bias.

ACKNOWLEDGEMENTS

C. R. is supported by VisualAI EP/T028572/1 and ERC-UNION-CoG-101001212.

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

# A    APPENDIX

## A.1    PSEUDO-CODE FOR REPRODUCIBILITY OF COSINE SCHEDULE

---
**Algorithm 1** Cosine Schedule

---
**Require:**  period $T \geq 0, \tau_- = 0.1, \tau_+ = 1.0$
   $ep \leftarrow$ current epoch
   $tau \leftarrow (\tau_+ - \tau_-) \times (1+\text{np.cos}(2\times\text{np.pi}\times ep/T))/2 + \tau_-$

---

Insert algorithm 1 into your favourite contrastive learning framework to check it out!

## A.2    IMPLEMENTATION DETAILS

**Evaluation details.** Following Jiang et al. (2021), we separate 5000 images for CIFAR10/100-LT as a validation set for each split. As we discussed in the main paper, the performance of the model depends on the relative position within a period $T$. Therefore we utilise the validation split to choose a checkpoint for further testing on the standard test splits for CIFAR10/100-LT. Precisely, for each dataset, we select the evaluation epoch for the checkpoint based only on the validation set of the first random split; the other splits of the same dataset are evaluated using the same number of epochs. Note that for ImageNet100-LT there is no validation split and we select the last checkpoint as in Jiang et al. (2021). For a fair comparison, we also reproduce the numbers from Jiang et al. (2021) in the same way.

**Division into head, mid, and tail classes.** Following Jiang et al. (2021), we divide all the classes into three categories: head classes are with the most number of samples, tail classes are with the least number of samples and mid are the rest. In particular, for CIFAR10-LT for each split there are 4 head classes, 3 mid classes, and 3 tail classes; for CIFAR100-LT there are 34 head classes, 33 mid classes, 33 tail classes; for ImageNet100-LT head classes are classes with more than 100 instances, tail classes have less than 20 instances per class, and mid are the rest.

## A.3    EXTENDED RESULTS

**Extension of Fig. 3** In Fig. 5 we provide full results of kNN accuracy on CIFAR10 when the model is trained with different fixed $\tau$ values and with coarse binary supervision. Especially tail classes are improved by instance discrimination (small $\tau_{\text{tail}}$).

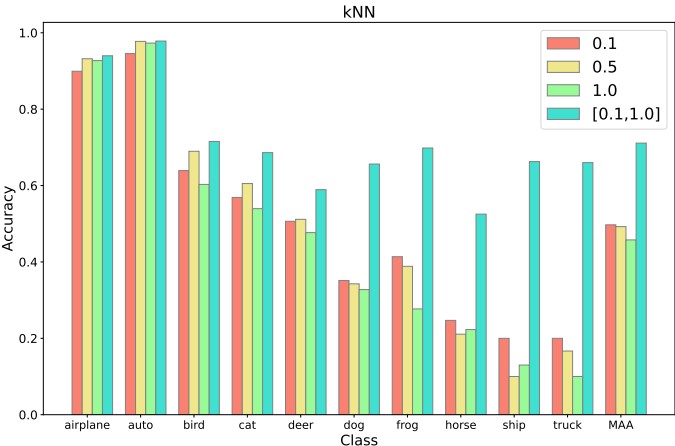

Figure 5: kNN accuracy for CIFAR10-LT trained with MoCo. Comparison between $\tau = 0.1, \tau = 0.5, \tau = 1.0$. [0.1, 1.0] denotes coarse binary supervision with $\tau_{\text{head}} = 1.0$ and $\tau_{\text{tail}} = 0.1$. MAA: mean average accuracy over all classes.

**Head-mid-tail classes evaluation.** In the following, we present a detailed comparison of SimCLR and SimCLR+TS on head, mid, and tail classes on CIFAR10-LT in Tab. 7, on CIFAR100-LT in Tab. 8 and on ImageNet100-LT in Tab. 9. We observe consistent improvement for all evaluation metrics for all types of classes over the three datasets.

| | CIFAR-10-LT | | | | | | | | | | | |
|---|---|---|---|---|---|---|---|---|---|---|---|---|
| | | | kNN@1 | | | | | | kNN@10 | | | |
| method | Head | | Mid | | Tail | | Head | | Mid | | Tail | |
| SimCLR | 84.93 | ± 3.44 | 54.08 | ± 4.24 | 32.14 | ± 7.44 | 88.03 | ± 3.32 | 53.76 | ± 4.80 | 29.52 | ± 9.44 |
| SimCLR + TS | 87.24 | ± 3.05 | 58.96 | ± 5.21 | 35.02 | ± 8.27 | 89.92 | ± 2.97 | 59.31 | ± 4.69 | 30.51 | ± 12.38 |
| | | | FS LP | | | | | | LT LP | | | |
| method | Head | | Mid | | Tail | | Head | | Mid | | Tail | |
| SimCLR | 76.38 | ± 5.24 | 63.20 | ± 2.95 | 62.60 | ± 3.63 | 89.52 | ± 3.15 | 56.98 | ± 4.74 | 29.88 | ± 8.11 |
| SimCLR + TS | 80.54 | ± 5.02 | 66.50 | ± 4.38 | 65.67 | ± 4.07 | 91.73 | ± 2.49 | 62.09 | ± 4.21 | 32.38 | ± 9.23 |

Table 7: Detailed evaluation on CIFAR10-LT. Evaluation metrics: kNN@1,10, FS LP states for few-shot linear probe, and LT LP states for long-tail linear probe. We report the average performance with the standard deviation over three different random splits for different sets of classes: head, mid, and tail.

| | CIFAR-100-LT | | | | | | | | | | | |
|---|---|---|---|---|---|---|---|---|---|---|---|---|
| | | | kNN@1 | | | | | | kNN@10 | | | |
| method | Head | | Mid | | Tail | | Head | | Mid | | Tail | |
| SimCLR | 53.87 | ± 2.12 | 24.56 | ± 1.51 | 7.26 | ± 0.39 | 58.46 | ± 1.79 | 22.15 | ± 1.47 | 2.83 | ± 0.61 |
| SimCLR + TS | 57.14 | ± 1.95 | 26.00 | ± 1.20 | 8.31 | ± 0.57 | 61.93 | ± 1.88 | 24.22 | ± 2.23 | 3.05 | ± 0.54 |
| | | | FS LP | | | | | | LT LP | | | |
| method | Head | | Mid | | Tail | | Head | | Mid | | Tail | |
| SimCLR | 33.48 | ± 1.24 | 24.25 | ± 2.12 | 19.12 | ± 1.35 | 62.19 | ± 1.80 | 26.56 | ± 1.46 | 3.92 | ± 0.46 |
| SimCLR + TS | 37.5 | ± 1.33 | 27.64 | ± 1.95 | 21.26 | ± 0.66 | 65.24 | ± 2.04 | 29.20 | ± 1.48 | 4.42 | ± 0.26 |

Table 8: Detailed evaluation on CIFAR100-LT. Evaluation metrics: kNN@1,10, FS LP states for few-shot linear probe, and LT LP states for long-tail linear probe. We report the average performance with the standard deviation over three different random splits for different sets of classes: head, mid, and tail.

| | ImageNet100-LT | | | | | |
|---|---|---|---|---|---|---|
| | | kNN@1 | | | kNN@10 | |
| method | Head | Mid | Tail | Head | Mid | Tail |
| SimCLR | 55.13 | 30.00 | 10.71 | 58.51 | 29.70 | 8.71 |
| SimCLR + TS | 57.23 | 30.26 | 13.14 | 60.41 | 29.53 | 10.14 |
| | | FS LP | | | LT LP | |
| method | Head | Mid | Tail | Head | Mid | Tail |
| SimCLR | 51.79 | 36.77 | 30.29 | 67.59 | 36.47 | 9.43 |
| SimCLR + TS | 60.41 | 40.38 | 33.57 | 70.67 | 38.85 | 10.29 |

Table 9: Detailed evaluation on ImageNet100-LT. Evaluation metrics: kNN@1,10, FS LP states for few-shot linear probe, and LT LP states for long-tail linear probe. We report the average performance for different sets of classes: head, mid, and tail.

**Influence of TS on Uniform vs Long-Tailed Distributions.** To further corroborate that TS particularly helpful for imbalanced data, we apply TS for the uniformly distributed data. In Tab. 10, we can observe that the cosine schedule yields significant and consistent gains for the long-tailed version of CIFAR10 (CIFAR10-LT), but not for the uniform one (CIFAR10-Uniform). We assume that both head classes and tail classes for long-tail distribution should be expected to benefit from a better separation between the two: on the one hand, the tail classes form better clusters and are thus easier to classify based on their neighbours, on the other hand, the clusters of the head classes are 'purified', which should similarly improve performance. Weather, for the uniform distribution, we do not observe such influence of TS and the performance changes only marginally.

| method | CIFAR10-Uniform | | | | CIFAR10-LT | | | |
|---|---|---|---|---|---|---|---|---|
| | kNN@1 | kNN@10 | FS LP | LT LP | kNN@1 | kNN@10 | FS LP | LT LP |
| MoCo | 83.47 | 84.87 | **90.19** | **87.70** | 63.00 | 64.10 | 68.89 | 63.99 |
| MoCo + TS | **83.78** | **85.85** | 90.02 | 87.40 | **65.68** | **65.91** | **72.31** | **66.64** |

Table 10: **Influence of TS on uniform vs long-tailed distribution.** Comparison of MoCo vs MoCo+TS on CIFAR10-Uniform and CIFAR-LT-imb100, one split. Evaluation metrics: kNN classifier, FS LP denotes few-shot linear probe, LT LP denotes long-tail linear probe.

### A.4 INFLUENCE OF THE POSITIVE SAMPLES ON CONTRASTIVE LEARNING

In Sec. 3.2, we particularly focused on the impact of the *negative samples* on the learning dynamics under the contrastive objective, as they likely are the driving factor with respect to the semantic structure. In fact, we find that the positive samples should have an inverse relation with the temperature $\tau$ and thus cannot explain the observed learning dynamics, as we discuss in the following.

To understand the impact of the *positive samples*, first note their role in the loss (same as Eq. (4)):

$$\mathcal{L}_{\mathrm{c}}^{i} = \log\left(1 + c_{ii}S_i\right) . \tag{6}$$

In particular, $c_{ii}$ scales the entire sum $S_i = \sum_{j \neq i} \exp(-d_{ij})$. As such, encoding two augmentations of the same instance at a large distance is much more 'costly' for the model than encoding two different samples close to each other, as each and every summand $S_i$ is amplified by the corresponding $c_{ii}$. As a result, the model will be biased to 'err on the safe side' and become invariant to the augmentations, which has been one of the main motivations for introducing augmentations in contrastive learning in the first place, cf. Tian et al. (2020b); Chen et al. (2020a); Caron et al. (2020).

Consequently, the positive samples, of course, also influence the forming of clusters in the embedding space as they induce invariance with respect to augmentations. Note, however, that this does not contradict our analysis regarding the impact of negative samples, but rather corroborates it.

In particular, $c_{ii}$ biases the model to become invariant to the applied augmentations for all values of $\tau$; in fact, for small $\tau$, this invariance is even emphasised as $c_{ii}$ increases for small $\tau$ and the influence of the negatives is diminished. Hence, if the augmentations were the main factor in inducing semantic structure in the embedding space, $\tau$ should have the opposite effect of the one we and many others (Wang & Liu, 2021; Zhang et al., 2022; 2021) observe.

Thus, instead of inducing semantic structure on their own, we believe the positive samples to rather play a critical role in influencing which features the model can rely on for grouping samples in the embedding space; for a detailed discussion of this phenomenon, see also Chen et al. (2021).

