# OpenReview forum: "Temperature Schedules for self-supervised contrastive methods on long-tail data"
_ICLR.cc/2023/Conference — ICLR 2023 poster_

### Official Review · Reviewer_bPFj · 2022-10-25

**Confidence:** 4
**Correctness:** 3
**Technical Novelty And Significance:** 2
**Empirical Novelty And Significance:** 3
**Recommendation:** 8

**Clarity, Quality, Novelty And Reproducibility:**

The paper is written clearly. The proposed method is not new but effective. The authors are suggested to include some clarifications on the experimental settings, as discussed in [W2].

**Strength And Weaknesses:**

Strength

**[S1]** Topic: the paper investigates self-supervised learning for long-tail data. This is one of the first steps toward applying self-supervised approaches to uncurated data, which is of great interest to the community;

**[S2]** Insight: the paper provides an empirical analysis of the impact of the temperature parameter on the head and tail categories;

**[S3]** Technology: the proposed dynamic temperature scheduling is technically sound;

**[S4]** Performance: compared to the vanilla contrastive learning and the SDCLR baseline, the proposed method demonstrates better performance on three long-tail benchmarks: CIFAR10-LT, CIFAR100-LT, ImageNet100-LT.

**[S5]** Presentation: the writing is clear.

---

Weaknesses

**[W1]** Interpretation:

i) I find it difficult to interpret the results of TS. The study in Sec.3.3 and Fig. 3 seems to indicate that TS achieves a trade-off between optimizing the performances of the head or tail classes: as shown in Fig.3, TS slightly reduces the performance of the head classes but significantly improves the performance of the tail classes. On the other hand, results in table 7/8/9 show that TS consistently improves the performance of both the head and tail classes. The improvements on the head and tail classes are also comparable: +2.1%/+2.4%  kNN@1 for the head/tail classes of ImageNet100-LT (table 9). It looks like TS is more than a simple trade-off?

ii) it is also unclear to me why decreasing the temperature during the training is the optimal scheduling solution (table 4 and 5). It looks like TS learns the head classes better at the early stage of the training then progressively prioritizes the tail classes. I wonder if TS may hurt the performance of the head classes at the later stage of the training. I would also be curious why alternative schedules perform worse than TS.

Overall, I’m happy to learn about the good performance of TS but the interpretation of TS deserves more investigation than what is presented in the current paper.

**[W2]** Evaluation. The results of SDCLR in table 3 look different from what were reported in the [SDCLR](https://arxiv.org/abs/2106.02990) paper. I wonder if these results are reproduced? If this is the case, I wonder if there is any difference in the experimental setting compared to the SDCLR paper. For example, the SDCLR paper reported results with 500 epochs pretraining while TS is trained with 800 epochs (Sec. 4.1), I wonder if different methods shown in table 3 share the same experimental configurations

**[W3]** The analysis in Sec. 3.2 and 3.3 is limited to contrastive learning. It is not clear to me if TS could be beneficial to non-contrastive SSL methods that use a temperature parameter, e.g. SwAV, DINO, etc.


**Summary Of The Paper:**

The paper explores self-supervised contrastive learning for long-tail data. It presents an empirical study of the impact of the temperature parameter on the head and tail categories. Based on the study, the paper proposes a simple yet effective solution to improve the imbalance pretraining which involves a dynamic temperature schedule. Compared to the vanilla contrastive learning and the SDCLR baseline, the proposed method demonstrates better performance on three long-tail benchmarks: CIFAR10-LT, CIFAR100-LT, ImageNet100-LT.

**Summary Of The Review:**

The paper provides an empirical analysis of the impact of the temperature parameters on self-supervised contrastive long-tail pretraining. A simple yet effective temperature scheduling solution is also proposed and demonstrates superior performance on three long-tail benchmarks. In general, I'm happy to learn about the good performance of this simple solution but would prefer some clarifications on the evaluation ([W2]). The paper could be stronger if it includes further investigations to interpret the results of TS ([W1]) and analysis of non-contrastive self-supervised methods ([W3]).

***
After rebuttal:

My concerns are addressed in the rebuttal. I thereby raise the score to 8.

---

> ### Author Response · Authors · 2022-11-16
> **Response to Reviewer bPFj — 2/2**
>
>
> **Does TS hurt the head classes at the end of training?**
>
> As discussed, the temperature oscillates multiple times and is therefore not biased against the head classes; in fact, as we show in the following table (CIFAR10-LT, imb100, one split), we find that most of the classes benefit from the proposed schedule:
>
> |        | Head 1   | Head 2   | Head 3   | Head 4   | Mid 1    | Mid 2    | Mid 3    | Tail 1   | Tail 2   | Tail 3  |
> | ------ | -------- | -------- | -------- | -------- | -------- | -------- | -------- | -------- | -------- | ------- |
> | fixed  | 86.1     | 96.0     | 80.7     | 65.2     | 72.0     | 75.5     | 21.6     | 35.7     | 29.6     | **2.8**     |
> | cosine | **89.6** | **96.4** | **87.7** | **74.3** | **74.7** | **83.9** | **27.7** | **51.0** | **38.8** | 2.6 |
>
>
> **Why do the alternative schedules perform worse?**
>
> We attribute the higher performance observed for cyclic cosine (and cyclic linear schedules) to the smoother transitions between the 'tasks', which avoid sudden spikes in the loss function and thus result in a more stable training behaviour. Following the suggestion of Fp8W reviewer, We additionally evaluate cyclic linear oscillation schedule that leads to similar performance as cyclic cosine function:
>
> | schedule         | FS LP |
> | ---------------- | ----- |
> | fixed            | 68.89 |
> | step             | 70.18 |
> | rand             | 70.26 |
> | lin.oscillations | 71.50 |
> | cosine           | 72.32 |
>
>
> > **[W2] Evaluation. The results of SDCLR in table 3 look different from what were reported in the SDCLR (https://arxiv.org/abs/2106.02990) paper. I wonder if these results are reproduced? If this is the case, I wonder if there is any difference in the experimental setting compared to the SDCLR paper. For example, the SDCLR paper reported results with 500 epochs pretraining while TS is trained with 800 epochs (Sec. 4.1), I wonder if different methods shown in table 3 share the same experimental configurations**
>
> All experiments in Table 3 share the same experimental configurations, hyperparameters and training environment. We use the code of SDCLR and even slightly improved their results by using feature l2-normalization (e.g. FS LP for CIFAR10 70.47 vs 71.03, for CIFAR100 25.27 vs 27.28)
>
> For ImageNet100-LT, the authors' implementation reproduces lower numbers than reported in the paper when training for 500 epochs (FS LP 39.42 code vs. 42.38 paper). To achieve comparable results as in the paper, we extend the training to 800 epochs for all methods on ImageNet100-LT. We run SimCLR and SDCLR based on the scripts provided by the authors and modify them correspondingly for SimCLR+TS.
>
> > **[W3] The analysis in Sec. 3.2 and 3.3 is limited to contrastive learning. It is not clear to me if TS could be beneficial to non-contrastive SSL methods that use a temperature parameter, e.g. SwAV, DINO, etc.**
>
> We fully agree with the reviewer that it would be interesting to further extend our proposed method to self-supervised learning schemes beyond contrastive learning, such as SwAV[1] and DINO[2] which use temperature within a knowledge distillation framework and thus do not rely on negatives during training. However, in our work we explicitly analyse how the impact of negatives changes with the temperature parameter. As such, an extension of our study to [1-2] is non-trivial, but we will include a discussion of SwAV and DINO in our manuscript, leaving a detailed analysis of their performance on LT data for future work.
>
> Refs:
>
> [1] Mathilde Caron, Ishan Misra, Julien Mairal, Priya Goyal, Piotr Bojanowski and Armand Joulin.  "Unsupervised learning of visual features by contrasting cluster assignments." In NeurIPS, 2021
>
> [2] Caron, Mathilde, HugoTouvron, Ishan Misra, Herve  Jegou,  Julien Mairal, Piotr Bojanowski and Armand Joulin. "Emerging properties in self-supervised vision transformers." In ICCV, 2021
>
> [3] Zixin Wen and Yuanzhi Li. “Toward understanding the feature learning process of self-supervised contrastive learning.” In ICML, 2021
>
> [4] Joshua David Robinson, Li Sun, Ke Yu, Kayhan Batmanghelich, Stefanie Jegelka, and Suvrit Sra. “Can contrastive learning avoid shortcut solutions?” In NeurIPS, 2021
>
> [5] Feng Wang and Huaping Liu. “Understanding the behaviour of contrastive loss.” In CVPR, 2021.

---

> > ### Comment · Reviewer_bPFj · 2022-11-26
> > **Thank you for the clarification.**
> >
> > I'd like to thank the authors for the clarification. My concerns are addressed. I have updated the score. Good job!

---

> ### Author Response · Authors · 2022-11-16
> **Response to Reviewer bPFj — 1/2**
>
> We thank the reviewer for the time taken to provide detailed and constructive feedback to our work and appreciate that the reviewer acknowledges that our work is among the first to apply SSL approaches to uncurated data, which is of "great interest to the community". Further, we are encouraged to find the simplicity and effectiveness of our proposed method as well as our writing to be highlighted so positively.
>
> In the following, we would like to address the reviewer's remaining concerns.
>
>
>
> > **[W1.1] I find it difficult to interpret the results of TS. The study in Sec.3.3 and Fig. 3 seems to indicate that TS achieves a trade-off between optimizing the performances of the head or tail classes: as shown in Fig.3, TS slightly reduces the performance of the head classes but significantly improves the performance of the tail classes. On the other hand, results in table 7/8/9 show that TS consistently improves the performance of both the head and tail classes. The improvements on the head and tail classes are also comparable: +2.1%/+2.4% kNN@1 for the head/tail classes of ImageNet100-LT (table 9). It looks like TS is more than a simple trade-off?**
>
> Based on our experimental results, we do not find that TS trades off accuracy on the tail classes against accuracy on the head classes, as we clarify in the following. In fact, our method improves the performance for all groups: head, mid, and tail classes.
>
> **First**, please note that Fig. 3 shows the results of a single head class (red) and a single tail class (blue); we will make this clear in the updated paper. For completeness, in the following table, we show the averages over all head and all tail classes:
>
> |                               | Head (4 classes) | Mid (3 classes) | Tail (3 classes) |
> | ----------------------------- | ---------------- | --------------- | ---------------- |
> | 0.1                           | 76.32            | 42.38           | 21.56            |
> | 0.5                           | 80.11            | 41.43           | 15.92            |
> | 1.0                           | 76.07            | 36.04           | 15.10            |
> | Coarse Supervision [0.1, 1.0] | 82.99            | 64.79           | 61.59            |
>
> It can indeed be seen that the head classes also benefit significantly from the coarse supervision, similar to the results observed by the reviewer fro TS (i.e., Tables 7,8, and 9).  However, please note that Figure 3 shows a setting different from TS and is thus not directly comparable.
>
> **Second**, we do not imply that the temperature scheduling necessarily trades off accuracy on the head classes to obtain better results on the tail classes. In fact, TS is not explicitly biased towards tail classes as it oscillates between high temperatures (+head classes) and low temperatures (+tail classes). When describing a trade-off between learning "easily separable features" and learning "instance-specific features", we refer to the features the model learns during training. TS is designed such that the model is biased to encode both 'easy' and 'hard' features, which we show to be particularly beneficial in the case of long-tailed data distributions (see also the answer to W2 of reviewer Fp8W).
>
> **Finally**, note that both head classes and tail classes should be expected to benefit from a better separation between the two: on the one hand, the tail classes form better clusters and are thus easier to classify based on their neighbours, on the other hand, the clusters of the head classes are 'purified', which should similarly improve performance.
>
>
>  > **[W1.2] It is also unclear to me why decreasing the temperature during the training is the optimal scheduling solution (table 4 and 5). It looks like TS learns the head classes better at the early stage of the training then progressively prioritizes the tail classes. I wonder if TS may hurt the performance of the head classes at the later stage of the training. I would also be curious why alternative schedules perform worse than TS.**
>
> **Cyclic Cosine Schedule**
>
> Please note that the temperature in our case is not simply decayed during training, it actually oscillates and completes multiple full cycles; see, e.g., the graph on the right in Table 5. As such, our proposed cosine schedule is different from the commonly employed cosine schedule used to decay the learning rate over the course of training, which typically uses only half a cosine cycle and is thus monotonously decreasing.
> In general, we find that this way of "changing tasks", i.e., switching between higher and lower values for tau, is beneficial for learning on long-tail data.
> To improve clarity, we propose to refer to our scheduling as _cyclic_ cosine schedule to clearly differentiate from the common cosine learning rate decay.

---

### Official Review · Reviewer_RFLZ · 2022-10-25

**Confidence:** 4
**Correctness:** 3
**Technical Novelty And Significance:** 2
**Empirical Novelty And Significance:** 2
**Recommendation:** 6

**Clarity, Quality, Novelty And Reproducibility:**

The overall quality is good, which is easy to follow and clearly written. Also the proposed method is novel.

**Strength And Weaknesses:**

Strength:
- The proposed method is interesting and is of a unique view.
- The proposed method requires no computation overhead, which is a tempting merit.

Weaknesses:
- The evaluation is not comprehensive. Only contrastive learning based methods are compared against.
- The improvement is quite limited compared with other long-tailed learning methods.

**Summary Of The Paper:**

This paper tackles the long-tailed learning problem by varying softmax temperature during the course of training. The authors argued that different magnitudes of the temperature value actually induce different learning preferences, which makes it possible to gradually switch the learning objective by carefully setting up a temperature scheduler. The proposed method has been evaluted on serveral long-tailed learning benchmarks, demonstrated its effectiveness.

**Summary Of The Review:**

An interesting method, yet with limited empirical improvments.

---

> ### Author Response · Authors · 2022-11-08
> **We would kindly ask for more clarifications, if possible**
>
> We thank reviewer RFLZ for their time and appreciate that RFLZ finds our method “interesting” and “of a unique view”. In order to adequately address the reviewer’s concerns, we would be grateful if RFLZ could elaborate and provide additional details regarding the potential weaknesses of our submission.
>
> W1 “The evaluation is not comprehensive. Only contrastive learning based methods are compared against.”
> If the reviewer could provide relevant references or suggestions regarding other methods to compare to that we might have missed, we are happy to provide additional comparisons to and discussions of those methods.
>
> W2 "The improvement is quite limited compared with other long-tailed learning methods."
> Further, we would like to respectfully rebut the assertion that the improvements are quite limited compared to other long-tailed learning methods. In fact, despite our method’s low computational overhead (as appreciated by the reviewer), we find consistent and significant improvements with respect to the current state of the art in self-supervised learning (SSL) on long-tailed (LT) data, namely SDCLR. To the best of our knowledge, we have included and discussed all relevant work. In case we did miss other work, we would highly appreciate if the reviewer could provide the respective references.

---

> > ### Comment · Reviewer_RFLZ · 2022-11-21
> > **Reply to Paper1728 Authors**
> >
> > Very sorry for this late reply. I understand that the scope of this paper is contrastive learning without knowing the label information. I would appreciate it if evaluations could be conducted on supervised contrastive learning as well, and to see how the proposed temperature scheduling interacts with or compared against other common long-tailed learning techniques in the supervised scenario.

---

> > > ### Author Response · Authors · 2022-11-24
> > > **Response to Reviewer RFLZ**
> > >
> > > We thank the reviewer for clarifying and report the results of the suggested experiments in the following; we are happy to add those to the supplement of the final paper if necessary.
> > >
> > > First, however, we would like to point out that the premise and goal of fully supervised learning on long-tailed (LT) data are very different from unsupervised learning on LT data. As such, the approaches and results are not directly comparable. In particular, the question we address in this work is **how to learn good representations in an unsupervised manner on LT data, without knowing the data data distribution**. In contrast, methods for supervised learning on LT data investigate **how to account for a known imbalance in the data distribution**, for example by applying re-sampling[3,4] or re-weighting[5,6] techniques.
> > >
> > > ### Supervised Contrastive Learning with Temperature Scheduling
> > >
> > > In the table below, we evaluate supervised contrastive learning (SCL) [1] with and without temperature scheduling (TS). In particular, we show the linear probing (LP) accuracy for models trained with supervised contrastive learning (SCL) [cite] on CIFAR10-LT with an imbalance ratio of 100; note that in comparison to the current state of the art, SCL [1] on its own is known to underperform on long-tailed data due to the dominance of the head classes and explicit re-balancing techniques are required to overcome this [2].
> > >
> > > | Method   | Linear probing (%) |
> > > | -------- | :----------------: |
> > > | SCL      | 72.42              |
> > > | SCL + TS | 72.79              |
> > >
> > > Consistent with our analysis, we find that TS does not yield significant improvements in this setting. In particular, please note that TS is an unsupervised technique designed to better separate head and tail classes if the respective labels are not known. Since full supervision already induces such a separation, we do not expect additional benefits from TS; in fact, full supervision can be regarded as an upper bound for TS.
> > > Hence, as our approach is not designed to improve the performance for supervised contrastive learning, we believe it should also not be compared to work in this field.
> > >
> > > Finally, as we find that SCL+TS performs comparably to SCL on its own, we would expect the resulting models to benefit to a similar degree from common long-tailed learning techniques, as for example reported in [2,7].
> > >
> > > Refs:
> > >
> > > ​​[1] Prannay Khosla, Piotr Teterwak, Chen Wang, Aaron Sarna, Yonglong Tian, Phillip Isola, Aaron Maschinot, Ce Liu, and Dilip Krishnan. "Supervised contrastive learning." In NeurIPS, 2020
> > >
> > > [2] Tianhong  Li, Peng Cao, Yuan Yuan, Lijie Fan, Yuzhe Yang, Rogerio S. Feris, Piotr Indyk, and Dina Katabi. "Targeted supervised contrastive learning for long-tailed recognition." In CVPR, 2022
> > >
> > > [3] Shin Ando and Chun Yuan Huang. “Deep over-sampling framework for classifying imbalanced data.” In Joint European Conference on Machine Learning and Knowledge Discovery in Databases, 2017
> > >
> > > [4] Mateusz Buda, Atsuto Maki, and Maciej A Mazurowski. “A systematic study of the class imbalance problem in convolutional neural networks.” Neural Networks, 2018.
> > >
> > > [5] Jonathon Byrd and Zachary Lipton. “What is the effect of importance weighting in deep learning?” In ICML, 2019
> > >
> > > [6] Salman Khan, Munawar Hayat, Syed Waqas Zamir, Jianbing Shen, and Ling Shao. “Striking the right balance with uncertainty.” In CVPR, 2019
> > >
> > > [7] Bingyi Kang, Yu Li, Sa Xie, Zehuan Yuan, and Jiashi Feng.”Exploring balanced feature spaces for representation learning.” In ICLR, 2020

---

> > > > ### Comment · Reviewer_RFLZ · 2022-11-25
> > > > **Reply to Paper1728 Authors**
> > > >
> > > > Thank you for the detailed response, which has addressed my concerns. I have updated the score. I hope the authors could take these suggestions as future works.

---

### Official Review · Reviewer_Fp8W · 2022-10-31

**Confidence:** 4
**Correctness:** 3
**Technical Novelty And Significance:** 3
**Empirical Novelty And Significance:** 3
**Recommendation:** 8

**Clarity, Quality, Novelty And Reproducibility:**

- The paper is clearly written. I especially appreciate that the main theoretical claim is explicitly laid out and tested.
- I have some small concerns about novelty, as described above.
- It seems like all relevant details are provided for reproducing the main results.

**Strength And Weaknesses:**

Strengths:
- The topic is interesting and offers a promising direction for deepening our theoretical understanding of contrastive learning.
- The work is clear and easy to follow with the main theoretical claim explicitly laid out and experiments set-up to test it directly.
- The experiments are conducted with a variety of models and datasets.


Weaknesses:
- It is unclear to me whether the cosine schedule is actually the optimal one. While it clearly outperforms the rapidly changing schedules like the step function, how does it compare, for example, to a linear (oscillating) schedule?
- The marginal performance gain from MoCo to MoCo+TS does not seem significantly higher in the imb 150 vs imb 50 case. It isn't entirely clear whether the results support the hypothesis posited initially by the authors, otherwise one might expect the marginal gain to be higher in the more imbalanced case. It would be useful to compare marginal performance gain on CIFAR10-LT and on CIFAR10 to confirm that the improvement from the temperature schedule is actually related to the data imbalance.
- There have been a few recent papers studying temperature in contrastive learning. In particular, how does your dynamic temperature scaling compare to the method in 'Dynamic Temperature Scaling in Contrastive Self-supervised Learning for Sensor-based Human Activity Recognition' (Khaertdinov et al., 2022)?

***Update after author response***
The authors have addressed these weaknesses to my satisfaction.


**Summary Of The Paper:**

Contrastive learning objectives are a popular and effective method for self-supervised learning. Most variations of the contrastive objective have a static temperature hyperparameter $\tau$. Previous work suggests that large values of $\tau$ lead to improved group-wise discrimination while small values lead to improved instance-wise discrimination. The authors study the effect of varying this temperature parameter in 'long-tail' settings with imbalanced data. They posit that long-tail class performance benefits from instance-wise discrimination ability and run initial 'coarse supervision' experiments to test this claim. Since this method requires (weak) labels, the authors also propose a dynamic temperature schedule that can be used in unsupervised settings. In a series of ablation experiments, they find that a cosine schedule that gradually switches between the two phases leads to the best performance.

**Summary Of The Review:**

I think the work is interesting and important, but have some concerns detailed in the Strengths and Weaknesses section.
I would be likely to raise my score and recommend acceptance if these concerns were addressed.

***Update after author response***
The authors have addressed these weaknesses to my satisfaction. I have raised my score accordingly.

---

> ### Author Response · Authors · 2022-11-16
> **Response to Reviewer Fp8W — 2/2**
>
>
> > **[W2] The marginal performance gain from MoCo to MoCo+TS does not seem significantly higher in the imb 150 vs imb 50 case. [...] It would be useful to compare marginal performance gain on CIFAR10-LT and on CIFAR10 to confirm that the improvement from the temperature schedule is actually related to the data imbalance.**
>
>
> We thank the reviewer for the suggestion and additionally ran experiments on CIFAR10 with a uniform distribution. We report the results in the following table:
>
> | Dataset            | schedule | KNN@1 | KNN@10 | FS LP | LT LP |
> | ------------------ | -------- | ----- | ------ | ----- | ----- |
> |                    | fixed    | 63.00 | 64.10  | 68.89 | 63.99 |
> | CIFAR10-LT imb 100 | cosine   | 65.68 | 65.91  | 72.31 | 66.64 |
> |                    | gain     | +2.68 | +1.81  | +3.42 | +2.65 |
> |                    | fixed    | 83.47 | 84.87  | 90.19 | 87.70 |
> | CIFAR10- Uniform   | cosine   | 83.78 | 85.85  | 90.02 | 87.40 |
> |                    | gain     | +0.31 | +0.98  | -0.17 | -0.30 |
>
> As can be seen, the cosine schedule yields significant and consistent gains for the long-tailed version of CIFAR10 (CIFAR10-LT), but not for the uniform one (CIFAR10- Uniform), which corroborates our hypothesis that TS is particularly helpful on imbalanced data.
>
> Moreover, regarding the results in Table 2 (MoCo and MoCo+TS), we would like to point out that results for the different imbalance ratios are not directly comparable, as a higher imbalance ratio makes the task of grouping tail classes inherently more difficult.
>
> In particular, please note that with an imbalance ratio of 50 (imb50), there are 100 samples for the smallest class, whereas with imb150, there are just 33. As a result, the kNN accuracy (kNN@10) for the tail classes drops significantly, as can be seen in the table below for 'fixed' schedule; specifically, in the table we report the results for kNN@10 for different sets of classes for CIFAR10-LT with an imbalance ratio of 50 and 150, when training the model with a fixed temperature.
>
> As the baseline performance for an imbalance ratio of 150 is thus much lower, the absolute gains are also lower. However, when looking at the the relative gains we indeed observe that the improvement increases along with the imbalance ratio, as can be seen in the following table
>
> |           	| imb 50 | imb50 | imb50  | imb150 | imb150 | imb150 |
> |---------------|--------|-------|--------|--------|--------|--------|
> |           	| head   | mid   | tail   | head   | mid	| tail   |
> | fixed     	| 95.22  | 67.40 | 36.03  | 94.67  | 55.27  | 15.60  |
> | cosine    	| 95.65  | 69.77 | 41.00  | 95.40  | 57.10  | 18.47  |
> | absolute gain | +0.43  | +2.37 | +4.97  | +0.73  | +1.83  | +2.87  |
> | relative gain (%) | +0.45  | +3.51 | +13.69 | +0.77  | +3.31  | +18.39 |
>
>
> We thank the reviewer for suggesting these additional experiments to support the hypothesis, which makes the results more comprehensible. We will update the paper with the discussion accordingly.
>
>
> > **[W3] There have been a few recent papers studying temperature in contrastive learning. In particular, how does your dynamic temperature scaling compare to the method in 'Dynamic Temperature Scaling in Contrastive Self-supervised Learning for Sensor-based Human Activity Recognition' (Khaertdinov et al., 2022)?**
>
> We thank the reviewer for pointing out the additional reference “Dynamic Temperature Scaling in Contrastive Self-supervised Learning for Sensor-based Human Activity Recognition”[1]. The motivation of [1] is to improve negative mining with the help of an additional pre-trained network and different temperatures for positive and negative samples. As such, [1] pursues a similar goal to [2] and [3], which we cite in our work; specifically, [2] propose dual temperature to control hardness-aware sensitivity for positives and negatives independently, and  [3] proposes an input-dependent temperature for uncertainty estimation for out-of-distribution detection. In contrast to [1-3], we analyse the underlying mechanics of contrastive learning and, based on this analysis, propose an input-independent temperature schedule to improve performance on long-tailed data. We will add this discussion to the paper.
>
> Refs:
>
> [1] Khaertdinov, Bulat, Stylianos Asteriadis, and Esam Ghaleb. "Dynamic Temperature Scaling in Contrastive Self-supervised Learning for Sensor-based Human Activity Recognition." IEEE Transactions on Biometrics, Behavior, and Identity Science (2022).
>
> [2] Chaoning Zhang, Kang Zhang, Trung X Pham, Axi Niu, Zhinan Qiao, Chang D Yoo, and In So Kweon. “Dual temperature helps contrastive learning without many negative samples: Towards understanding and simplifying moco.” In CVPR, 2022
>
> [3] Oliver Zhang, Mike Wu, Jasmine Bayrooti, and Noah Goodman. “Temperature as uncertainty in contrastive learning.” arXiv preprint arXiv:2110.04403, 2021

---

> > ### Comment · Reviewer_Fp8W · 2022-11-17
> > **Concerns are addressed**
> >
> > Thank you for addressing the concerns raised in my review. I know how stressful it is as an author to wait after the discussion period to see whether reviewers will even read your rebuttals, let alone agree with them and change scores accordingly. I'd like to spare you the suspense and have already updated my score.

---

> > > ### Author Response · Authors · 2022-11-17
> > > **Thank you!**
> > >
> > > We are truly grateful for the early feedback and the kind words, it is indeed very motivating.
> > >
> > > We would further again like to thank the reviewer for the detailed and highly constructive feedback, which helped us to improve and strengthen our submission.
> > >
> > > Kind regards,
> > > Authors of paper 1728

---

> ### Author Response · Authors · 2022-11-16
> **Response to Reviewer Fp8W — 1/2**
>
> We thank the reviewer for the time taken to provide detailed and constructive feedback to our work. We are encouraged to find that our work is considered a "promising direction" and appreciate that our paper is deemed "clear and easy to follow".
>
> In the following, we would like to address the reviewer's remaining concerns.
>
> > **[W1]  It is unclear to me whether the cosine schedule is actually the optimal one. While it clearly outperforms the rapidly changing schedules like the step function, how does it compare, for example, to a linear (oscillating) schedule?**
>
>
> While we empirically find the cosine schedule to yield the best performance among the tested schedules (random, step, fixed, cos, see Table 5), we fully agree with the reviewer that we cannot conclude from our experiments that it constitutes the best possible schedule — in fact, given the infinite space of potential schedules and dataset configurations, such a claim would be untenable.
>
> The finding of this paper is rather that 'task switching' in general can yield a better structured representation space in the context of long-tail distributed datasets, as the performance improves for all the tested schedules (step, rand, cos) over the baseline (fixed), see Table 5 and below.
>
>
> We appreciate the reviewer’s suggestion and additionally evaluated a linearly oscillating schedule, for which we observe similar performance gains as with the cosine function (see table below). We hypothesise that the higher performance observed for the cosine and the linear schedules might be due to smoother transitions between the 'tasks', which avoid sudden spikes in the loss function and might thus result in a more stable training behaviour; we will extend our discussion regarding this in the paper.
>
> | **Schedule**        | **FS LP** |
> | ------------------- | --------- |
> | fixed               | 68.89     |
> | step                | 70.18     |
> | rand                | 70.26     |
> | linear oscillations | 71.50     |
> | cosine              | 72.32     |

---

### Author Response · Authors · 2022-11-17
**Summary of Changes**

We thank the reviewers for their thoughtful comments and suggestions. In the following, we provide a brief summary of the changes made to the paper:

1. We include an additional alternative temperature schedule, namely the linear oscillation function. We update Table 5 correspondingly.
2. We include additional experiments for the uniformly distributed data in Appendix.
3. We update related work discussion with respect to negatives.
4. We clarify the setting for Figure 3 in the text and caption.

These changes improved the clarity, and additional experiments even further confirm our hypothesis.

---

### Public Comment · ~Zhihan_Zhou2 · 2023-02-13
**Congratulations to the nice work! Recommend our work on self-supervised long-tailed learning.**

Hi,

Congratulations to the nice work! Our recent ICML'22 paper BCL (https://proceedings.mlr.press/v162/zhou22l/zhou22l.pdf) also studied self-supervised long-tailed learning. Would you mind citing our work? Many thanks!

Best,

Zhihan

---

### Decision · Program_Chairs · 2023-01-20

**Decision:**

Accept: poster

**Justification For Why Not Higher Score:**

- Proposed method is not totally novel. Prior literature already studies that different softmax temperature value induce different learning focus, e.g. https://arxiv.org/abs/2007.07314, https://arxiv.org/abs/2103.01550, etc.
- Proposed method only brings improvement in the unsupervised setting and not on supervised setting, thereby limiting the audience of interest.

**Justification For Why Not Lower Score:**

The simplicity and effectiveness of the proposed method for unsupervised representation learning on long tail data as appreciated by all the reviewers.

**Metareview: Summary, Strengths And Weaknesses:**

The paper attempts to improve unsupervised representation learning for long tail data. In this regards, the author propose a very simple method of varying softmax temperature during the course of training. It is argued that different temperature regime value induce different learning focus: "group-wise discrimination" vs "instance discrimination". The proposed method has been evaluated on several long-tailed learning benchmarks, demonstrated its effectiveness. We thank the authors and reviewers for actively engaging in discussion and taking steps towards improving the paper including for providing additional experiments and improving writing clarity. All the reviewers appreciate the simplicity and effectiveness of the proposed method.

**Note From Pc:**

if the above contains the word "oral" or "spotlight" please see: "oral" presentation means -> notable-top-5% and "spotlight" means -> notable-top-25%. As stated in our emails, we are disassociating presentation type from AC recommendations

**Summary Of Ac-Reviewer Meeting:**

N/A